# Did a digital quality of life (QOL) assessment and practice support system in home health care improve the QOL of older adults living with life-limiting conditions and of their family caregivers? A mixed-methods pragmatic randomized controlled trial

Richard Sawatzky[1,2,3*©], Kara Schick-Makaroff[4©], Pamela A. Ratner[5], Jae-Yung Kwon[6], David G. T. Whitehurst[7], Joakim Öhlén[3,8], Alies Maybee[9], Kelli Stajduhar[6], Lisa Zetes-Zanatta[10], S. Robin Cohen[11,12]

**1** School of Nursing, Trinity Western University, Langley, British Columbia, Canada, **2** Centre for Advancing Health Outcomes, Providence Health Care Research Institute, Vancouver, British Columbia, Canada, **3** Institute of Health and Care Sciences, and Centre for Person-Centered Care (GPCC), Sahlgrenska Academy, University of Gothenburg, Gothenburg, Sweden, **4** Faculty of Nursing, University of Alberta, Edmonton, Alberta, Canada, **5** School of Nursing, University of British Columbia, Vancouver, British Columbia, Canada, **6** School of Nursing & Institute on Aging and Lifelong Health, University of Victoria, Victoria, British Columbia BC, Canada, **7** Faculty of Health Sciences, Simon Fraser University, Burnaby, British Columbia, Canada, **8** Palliative Centre, Sahlgrenska University Hospital Västra Götaland Region, Gothenburg, Sweden, **9** Independent patient partner, **10** Kamloops Community Programs and Surgical Services Network, Interior Health Authority, Kamloops, British Columbia, Canada, **11** Departments of Oncology and Medicine, McGill University, Montreal, Quebec, Canada, **12** Lady Davis Research Institute of the Jewish General Hospital, Montreal, Quebec, Canada

© Shared first authors who contributed equally to this project.
* rick.sawatzky@twu.ca

## Abstract

We aimed to improve the quality of life (QOL) of homecare patients (≥ 55 years of age) who had chronic life-limiting conditions and that of their family caregivers by making QOL assessment data available via a digital QOL and practice support system (QPSS). We hypothesized that access to QPSS data in home health care would result in improved QOL for patients or their family caregivers. We further sought to understand how to integrate the use of QOL information into home health care. Our mixed-methods study, including a pragmatic randomized controlled trial (PrCT; registered at ClinicalTrials.gov #NCT02940951), was conducted with nine home healthcare teams in Canadian urban areas. The qualitative research included focus groups and interviews with home healthcare teams (N = 118) to determine how to integrate the QPSS into their practice. Participating homecare patients were assigned to an intervention group (N = 166), where home healthcare teams had access to patients' and their family caregivers' QOL data via the QPSS, or a usual care group (N = 165). Where possible, one family caregiver per patient was recruited (intervention N = 62; usual care N = 51). Primary outcomes, measured every two months for one year,

**Data availability statement:** The data required for replicating the quantitative analyses are available via the following repository: Sawatzky R. Quality of Life Assessment and Practice Support System project: supporting documentation", https://doi.org/10.5683/SP3/A4JG2R, Borealis. Excerpts of the transcripts relevant to the qualitative analyses are provided within the paper.

**Funding:** 1) Canadian Institutes for Health Research: https://webapps.cihr-irsc.gc.ca/decisions/p/project_details.html?applId=334627&lang=en 2) Canada Research Chairs 3) Fraser Health Authority: https://www.fraserhealth.ca/ 4) Cambian: https://www.cambian.com/ 1) & 2) did not play any role in the study design, data collection and analysis, decision to published, or preparation of the manuscript. 3) facilitated data collection from patients and home healthcare providers, but had no other role. 4) facilitated data collection via their Collaborative Healthcare Information Services Platform, and had not other role.

**Competing interests:** The authors have declared that no competing interests exist.

were patients' and family caregivers' QOL trajectories. Longitudinal structural equation models were used to compare the trajectories. The home healthcare teams preferred to have QOL scores presented as tables and graphs, with family caregivers' data linked to each patient. Despite the enthusiasm expressed by the home healthcare teams, and efforts to satisfy their preferences, they infrequently accessed the QOL information. While we observed substantial individual-level variability in patients' and family caregivers' QOL trajectories, the average trajectories for the PrCT groups were similar. Making QOL assessment data available via a digital platform may not be sufficient to achieve measurable improvements for patients and family caregivers.

## Introduction

Older adults who have incurable, advancing life-limiting conditions often choose to be cared for at home and to live their last years or months as fully and as comfortably as possible. This requires health care that focuses on improving their quality of life (QOL) and the QOL of their family caregivers (i.e., those in supportive roles who undertake vital care work or emotional management). Such a focus requires early identification, assessment, and treatment of multiple complex symptoms and concerns in a range of life domains: physical, psychosocial, and spiritual [1]. Comprehensive, ongoing reporting of QOL and healthcare concerns, and action based on those reports by home healthcare teams, are required to inform patient- and family-centered approaches to care planning and decision making to improve the quality of care and to enhance QOL. Optimal care occurs when healthcare providers understand that concerns in one life domain impact other domains. For example, bodily pain can be increased by spiritual or psychological distress and *vice versa*, and the most effective pain management involves addressing these sources of distress along with interventions aimed directly at reducing pain [2].

Home healthcare teams (including nurses, nurse practitioners, social workers, physicians, and other health professionals) play a key role in providing care at home to people who have advancing, chronic life-limiting conditions. In this role, they aim to ensure that the full range of healthcare needs relevant to the QOL of their patients and that of their family caregivers are recognized and adequately addressed. This includes comprehensive assessments not only of patients' symptoms, but also other dimensions of their QOL (e.g., physical, psychological, social, and existential) [3,4]. In addition, routine assessments of family caregivers' QOL are required to prevent their own health and wellbeing from worsening (many are seniors with their own health issues) and to enable them to provide care both at home and for a longer duration, if they wish to do so. Family caregivers often work closely with the home healthcare teams and, together, assume major responsibility for the coordination and delivery of care [5–7]. Though normally done willingly [8], the work of family caregivers in caring for someone who has a life-limiting illness at home impacts multiple dimensions of their own QOL, and even leads to their increased mortality [9,10].

While there is a clear need to ensure that the QOL concerns and supportive needs of family caregivers are also routinely assessed and addressed by home healthcare teams, this aspect of care is often neglected [5,11].

QOL assessments can be facilitated by using standardized questionnaires, which are often referred to as measures of QOL or patient-reported outcome measures (PROMs). PROMs refer to self-report instruments used to obtain appraisals from healthcare recipients about outcomes relevant to their health and QOL [12]. Most PROMs are multidimensional in that they measure various life domains, including those related to symptoms, functional status, health status, and psychological, social, and spiritual wellbeing. The routine use of such QOL assessment instruments may enable healthcare professionals to efficiently assess and address the needs and concerns of their patients and fluctuations in their patients' QOL, including symptoms, functional status, psychological, social, and existential wellbeing [13–16].

Primary studies and systematic reviews suggest that providing healthcare professionals with QOL assessment data may improve patients' QOL, enhance clinician-patient communication, raise awareness of problems that would otherwise be unidentified, and support care planning and multidisciplinary collaboration [14,17–22]. Healthcare administrators and managers increasingly advocate for the routine use of PROMs because of their potential to improve patient care and to reduce costs by helping clinicians work more efficiently, enabling them to more quickly and accurately assess and address patients' problems as they arise [23]. In addition, information from PROMs can be used for program evaluation and quality improvement from a person-centered care point of view, as well as economic evaluation [24–26]. Although the use of similar information about the QOL and healthcare experiences of family caregivers has been less studied, it is reasonable to expect that this information would facilitate improvement of caregivers' wellbeing and potentially enhance their ability to continue providing care, especially given that they are typically reluctant to mention their own needs without prompting [27].

Despite the widely recognized need to pay attention to patients' and family caregivers' perspectives of their QOL and healthcare experiences, and the availability of many QOL assessment instruments (i.e., PROMs), the integration of these assessment instruments into the daily practice of home healthcare teams has been elusive. User-centered digital health systems, made accessible at the point of care, have been recommended to facilitate the integration of self-reported QOL assessment instruments in practice [28–31]. For the integration of such systems to be successful, it is imperative that end users (including healthcare professionals, patients and family caregivers) be closely involved in their design, development, implementation, evaluation, and modification [32]. The optimal system must be user-friendly, deliver relevant and timely access to information at the point of care, be well-integrated into established workflows, include mechanisms that facilitate care planning and decision-making processes, and be adaptable to the needs of all users. At the same time, the system must be integrated with other health information systems so that patient and family caregiver information can be used for program evaluation and health system performance monitoring.

Various digital QOL assessment tools have been studied [28,29,31,33–35]. However, the studies of people with life-limiting conditions have predominantly focused on oncology patients, and there has been little emphasis on family caregivers' QOL. There is also a lack of knowledge about how to best integrate such systems into the routine practice of home healthcare teams and whether such routine assessments will make a difference for homecare patients in terms of their QOL and quality of care.

We therefore aimed to improve the QOL of homecare patients who had chronic life-limiting conditions, and of their family caregivers, by making QOL assessment data available at the point of care via a digital health platform, herein referred to as the QPSS. We expected that the home healthcare teams would be engaged throughout the integration of the QPSS and would access QOL assessment data from the QPSS as part of their routine care. We hypothesized that the provision of access to the QPSS in home health care, over a period of one year (the intervention), would result in improved QOL, for patients or their family caregivers, over time (the outcome). This hypothesis was based on the implicit assumption that if a) patients or family caregivers completed QOL assessments, b) the included tools were those endorsed by the healthcare professionals, c) QOL assessment data were integrated into a digital health information system, d) the data were readily

available to the home healthcare teams, and e) the results were presented in accessible ways, then the home healthcare teams would use the information in their care, which would, in turn, improve patients' or family caregivers' QOL outcomes. To accomplish this, we first sought to understand how to best integrate QOL information into the daily practice of home healthcare teams.

## Methods

### Study design

We designed the study as a pragmatic trial (with an emphasis on a real-world population and setting [36]; see Fig 1) following a longitudinal convergent mixed-methods approach based on principles of action-oriented integrated knowledge translation (iKT) [37]. The study included two stages aligned with the Knowledge to Action Framework (see Fig 2). Stage 1 involved qualitative research with home healthcare teams to (i) adapt the QPSS to the local context and (ii) determine how to best integrate it into local practice. Stage 2 involved a triple-blind (participants, outcome assessors, and investigators) pragmatic randomized controlled trial (PrCT) [36], where homecare patients were randomly assigned to an intervention group or a control group (see Fig 3). For the patients in the intervention group, home healthcare teams were provided with access to the QOL data provided by participating patients and family caregivers, via the QPSS, in addition to providing their usual care. Patients in the control group received usual care (i.e., the home healthcare teams were not provided with access to patients' QOL data). Stage 2 also included qualitative research and the monitoring of the use of the QPSS by the home healthcare teams. Both stages followed patient-oriented research principles (e.g., meaningful patient engagement and application to real-world problems) [38] with a research team that included a Patient Advisory Committee, multi-disciplinary healthcare professionals, decision-makers, researchers, and representatives of non-profit health advocacy organizations. The PrCT was registered at ClinicalTrials.gov (#NCT02940951). Research ethics approval for the study was provided by the Trinity Western University Research Ethics Board (#15F15).

### Study intervention

The QPSS prototype, designed in a prior study, was guided by the Knowledge to Action Framework with extensive input from diverse knowledge users from palliative care settings across Canada (Action Cycle A: 'identify problem; identify, review, select knowledge') [39,40]. The QPSS was designed to facilitate healthcare providers' use of QOL assessment instruments by providing access to instantaneously scored results of item and domain scores, including trends over time, reported in customizable tabular and graphical formats. Patients and family caregivers in both the intervention and control groups were given dedicated accounts through which they could complete the QOL assessment instruments and view the results. Home healthcare teams could access the QPSS to view the QOL assessment results only for patients and family caregivers in the intervention group.

The selection of QOL assessment instruments was informed by the original QPSS prototype development and subsequent initial focus group meetings with the participating home healthcare teams. For patients, we included the Edmonton Symptom Assessment System - Revised (ESAS-r) [41] and the McGill Quality of Life Questionnaire – Expanded (MQOL-E) [42]. For family caregivers, we included the Carer Support Needs Assessment Tool (CSNAT) [43,44] and the Quality of Life in Life-Threatening Illness - Family Carer Version 3 (QOLLTI-F v3) [45] (see Table 1).

All participating patients and family caregivers were invited by the research staff to complete the QOL assessment instruments every two months for a period of one year from the time of enrolment, or until discharge or death. Following an invitation for study orientation and onboarding at the participant's home, the research staff implemented a standardized QOL assessment schedule, involving up to eight reminders for each assessment occasion, as well as four follow-up attempts to complete a subsequent assessment if a previous assessment was missed. The participants could complete the QOL assessments via any of the following modes: (i) logging into their QPSS account using their personal computer or tablet,

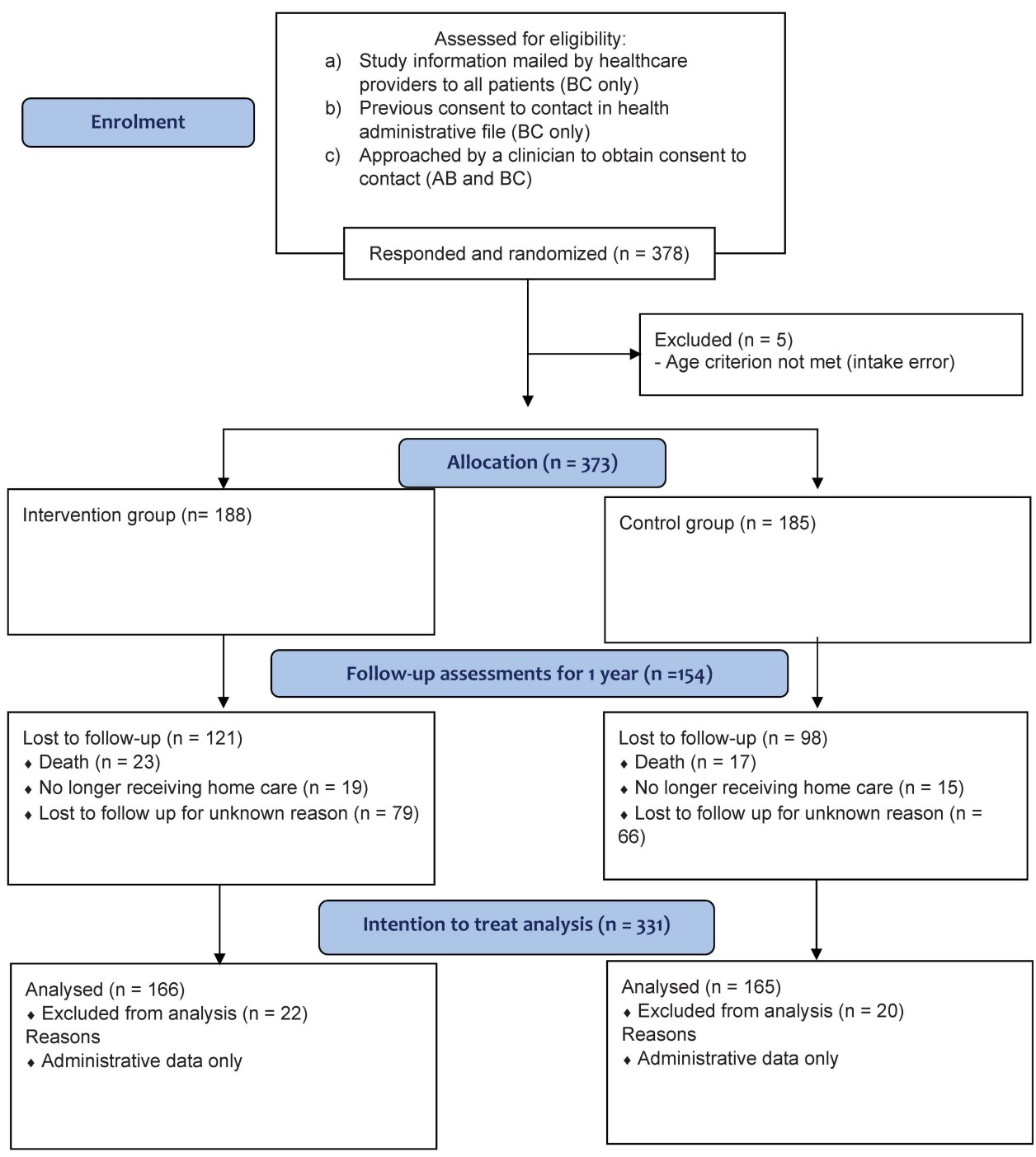

**Fig 1. CONSORT Flow Diagram.**

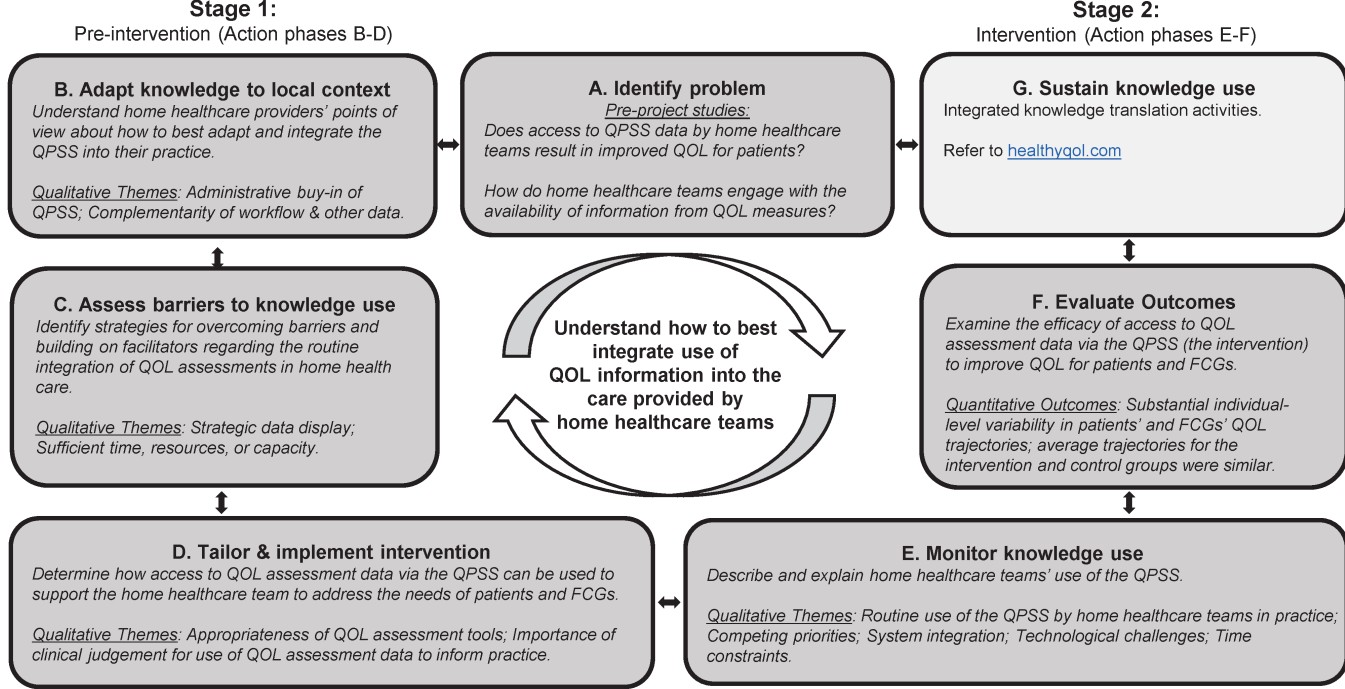

**Fig 2. Overview of Project Action Phases and Results. Based on the Knowledge to Action Cycle; A through G are "action cycles"[37]. FCG = family caregiver. QOL = quality of life. QPSS = quality of life and practice support system.**

(ii) completing the assessment instruments on paper and returning them by mail, (iii) completing the instruments with a research staff member during an in-person home visit, or (iv) having a research staff member administer the instruments over the phone. For the latter three modes, a research staff member entered the responses into the participant's QPSS account on their behalf.

**Study setting.** Our study was conducted with nine home healthcare teams servicing urban areas in two Canadian provinces, British Columbia (BC) and Alberta (AB). The seven teams in BC initially followed a case management model of care and during the study transitioned to a model of team-based patient-centered care with direct formal linkages to the primary care physician(s). The purpose of this change was to extend the reach of the primary care provider and allow virtual collaborative visits in the home to better support the clients' needs. Five of the seven sites in BC continued into Stage 2 of the project (the PrCT). The eighth home healthcare team involved a multidisciplinary community-based clinic that offers home health care for low-income seniors and included a nurse practitioner, and the ninth home healthcare team provided services by nurse practitioners exclusively. These two teams from Alberta were both part of Stage 2.

**Recruitment and inclusion criteria.** During Stage 1 of the study, multidisciplinary clinicians (registered nurses, licensed practical nurses, nurse practitioners, occupational therapists, physiotherapists, social workers, and administrators) from all nine home healthcare teams were invited to participate via information sessions and emails between January 2016 and July 2017; the start dates were staggered across the sites.

Recruitment of patients and family caregivers for the pragmatic randomized controlled trial (Stage 2) commenced on March 12, 2018, and discontinued on March 31, 2019, due to resource constraints. The study participants provided written or verbal consent.

Patients were approached for recruitment through several means: (i) if they had indicated at some time to the home-care service that they were willing to be contacted for research (general consent) (BC only), (ii) if they contacted the

**Table 1. QOL Assessment Instruments.**

| Instruments | Construct | Domains measured (# of items) | Response scales | Calculation of domain scores | Original purpose of the instrument | Use in this study |
|---|---|---|---|---|---|---|
| MQOL-E [42] | Patient's quality of life over the past two days | Physical (3); Psychological (4); Existential (4); Social (3); Environment (1); Cognition (2); Health care (2); Feeling like a burden (1) | 0 to 10 with item-specific anchors | Domain scores are the means of the corresponding items (reverse scoring of negatively worded items). A summary score is obtained based on the mean of the 8 domains. 0 (worst possible QOL) to 10 (best possible QOL) | To comprehensively measure QOL for people with life-threatening illness throughout their illness trajectory | Patient-completed QOL assessments provided to clinicians (intervention group only) and outcomes evaluation (both groups) |
| ESAS-R [41] | Patient's current symptoms | Symptoms (8); Wellbeing (1); Self-identified problem (1) | 0 (no symptom) to 10 (worst possible) | n/a | To support healthcare providers in monitoring a patient's condition and symptom severity [46] | Patient-completed QOL assessments provided to clinicians (intervention group only) |
| QOLLTI-F v3 [45] | Family caregiver's QOL over the past two days | Overall QOL (1); Environment (2); Patient condition (1); Caregiver's own condition (5); Caregiver's outlook (3); Quality of care (2); Relationships (2); Financial worries (1) | 0 to 10 with item-specific anchors | Domain scores are the means of the corresponding items (reverse scoring of negatively worded items). A summary score is obtained based on the mean of the 7 domains. 0 (worst possible QOL) to 10 (best possible QOL) | To better understand and assess QOL of family caregivers of people living with life-threatening illness (at all stages) for use in clinical, research, and teaching settings [47] | Family caregiver completed assessments of their own QOL provided to clinicians (intervention group only) and outcomes evaluation (both groups) |
| CSNAT [44] | Supportive needs of family caregivers | 14 items representing 14 broad domains of family caregiver's need for support | 0 (no need for support), 1 (a little more), 2 (quite a bit more), 3 (much more). | n/a | To help family caregivers identify key support needs to a healthcare provider [44] | Family care giver-identified needs provided to clinicians (intervention group only) |

n/a = not applicable.

research team expressing interest in the study in response to a general letter sent by the healthcare organization (BC only), or (iii) if they were agreeable to a referral to the study team by participating healthcare professionals (BC and AB). Eligible patients who had provided general consent to be contacted for research purposes were contacted by the research personnel with an invitation to participate. In addition, the participating healthcare professionals were asked to continuously screen their patients for eligibility and to ask whether they would agree to be contacted by research personnel. The research personnel explained the study and screened for the following inclusion criteria: 55 years of age or older; receiving ongoing home health care because of having one or more advancing life-limiting conditions; and able to speak English and provide written informed consent. Because the invitations to participate were shared through different mechanisms and numerous people, it was not possible to determine the number of patients initially screened.

The family caregivers were nominated by the enrolled patients. Inclusion criteria for family caregivers required them to be the person most involved in the care of the enrolled patient, an adult, able to speak English, and willing to provide written informed consent. A maximum of one caregiver per patient could be recruited. Patients could be included without a caregiver, but family caregivers were required to have a patient enrolled to be eligible, irrespective of the extent of data provided by the patient.

## Qualitative methods

Qualitative data from home healthcare teams were collected during both stages. In the pre-intervention stage (Stage 1), to inform Steps B-D of the Knowledge to Action Framework (Fig 2), we conducted 45 focus groups and six interviews with participating clinicians at all nine sites (April 2016 to April 2018). During the intervention stage (Stage 2), to inform Step E of the Knowledge to Action Framework, we conducted seven focus groups and five interviews at seven participating sites (March 2018 to March 2020) (see Table 2). Focus groups and interviews were conducted by research coordinators or the study's co-lead (KSM) with trainees (with no prior relationships) in private rooms at the home healthcare sites or by phone, lasting 30–90 minutes. Field notes, along with debriefing between interviewers and study leads, enhanced data quality and reflexivity. Semi-structured focus group/interview guides were iteratively revised with the research team, informed by ongoing analysis as described below. Focus groups and interviews were audio-recorded. Only those focus groups that included discussions about the research questions and objectives, and not the Stage 1 focus groups that predominantly entailed hands-on training, were transcribed. All Stage 2 recordings and all interviews were transcribed verbatim.

**Table 2. Number of Focus Groups and Individual Interview Participants.**

| Stage | Study sites | # of focus groups | # of unique focus group participants[a] | # of unique interview participants[a] | Total # of unique participants[a] |
|---|---|---|---|---|---|
| 1 | | | | | |
| | British Columbia[b] | 42 | 87 | 6 | 88 |
| | Alberta[c] | 3 | 12 | 0 | 12 |
| | Total | 45 | 99 | 6 | 100 |
| 2 | | | | | |
| | British Columbia[d] | 4 | 28 | 1 | 28 |
| | Alberta[c] | 3 | 10 | 4 | 10 |
| | Total | 7 | 38 | 5 | 38 |

Stage 1 = pre-intervention period. Stage 2 = Intervention period.

[a]Participants who were in more than one focus group or interview were counted once.

[b]Sites 1–7.

[c]Sites 8 and 9.

[d]Sites 2–6 (sites 1 and 7 were not included in Stage 2).

Informed by interpretive description [48], data were analyzed descriptively to tailor the implementation of the QPSS intervention (Stage 1) and to monitor the clinicians' use of the QPSS (Stage 2). Clinicians' frequency of accessing the QPSS during Stage 2 was tabulated to inform the emerging findings. Initial transcripts, recordings, and field notes were used to create a codebook, iteratively refined during the analysis. We first organized the data by the action phases (A to G, see Fig 2) of the Knowledge to Action Framework. Subsequently, we conducted analyses within each action phase and created themes that described how the clinicians engaged with the QPSS. Subsequent interpretive analyses included triangulation of the themes with the quantitative data about each clinician's use of the QPSS. The data were coded by two research coordinators under the supervision of KSM; coding inconsistencies were resolved through discussion to achieve consensus. The data were managed with NVivo™ and analyzed thematically. We ensured trustworthiness and rigor by incorporating notions of: (i) credibility, including iterative cycles of engagement at local sites; iterative team analysis discussions; triangulation; negative/alternative case analysis to show less-common perspectives; and multiple sources of data; (ii) confirmability, including audit trails and field notes; and (iii) transferability through detailed reporting of the context [48,49].

## Quantitative methods

**Sample size and randomization.** The determination of sample size for the PrCT, which relied on latent variable model analysis, was not straightforward; researchers typically rely on "rules of thumb" although there are some *a priori* analytical tools. Our sample size determination was guided by feasibility and *a priori* statistical power considerations. We assumed that the effect of the routine use of the QPSS, by the health professionals, would be equivalent to changing an "average" day into a "good day" for their patients, based on the original McGill Quality of Life Questionnaire [MQOL][50], which corresponds to a standardized mean difference of $d = .70$ and a $f$ value of.35 for a one-way analysis of variance. As a rough guide, for a two-group, two-tailed ANOVA, with power of.80, and alpha of.05, a total of 68 patients would be required to detect an effect size of $f = .35$, and a total of 128 patients would be needed to detect a smaller effect size of $f = .25$ (corresponding to a Cohen's $d$ of.50). Further, following the recommendations of L. Muthén and B. Muthén [51], a simulation-based power analysis on a linear growth structural equation model with seven equidistant measurement occasions (see the full model specification below) indicated sample sizes of 77 and 150 would be needed to detect effect sizes of $d = .70$ and $d = .50$, respectively, with power of.80 and alpha of.05, and variances of 0.25 and 0.09 for the intercept and slope, respectively, with no missing data. Samples of this size would provide sufficient power to reject the hypothesis that the mean of the slope (the rate of change over time) is zero. A larger sample size was sought, given that substantial loss to follow-up was anticipated due to the nature of the study population, and that there was little guidance to facilitate the adjustment necessary to accommodate loss to follow-up.

A freely available, online, block randomization tool was used (www.randomization.com). Three hundred sixty potential participants were randomized 1:1 in 36 blocks of 10. A co-author (SRC), not involved in recruitment or data collection, obtained the randomization sequence, placed a paper indicating group assignment into sequentially numbered opaque envelopes, and gave the envelopes to the research staff. Only the research coordinator and clinicians knew which patients and family caregivers were assigned to the intervention group. Patients and family caregivers were not informed about group assignment, although those in the intervention group may have indirectly learnt about it from their health professionals. All patients and family caregivers were told that their health professionals may or may not have seen their completed questionnaires. This was intended to reduce the risk that patients and family caregivers assumed that their healthcare providers were aware of any problems highlighted in their questionnaire responses. This was important for both groups, since we did not know the extent to which the clinicians would actually access the completed questionnaire scores of those assigned to the intervention group.

**Quantitative data collection.** Questionnaire data were collected via the same procedures as described above for the study intervention. Demographics (patients and family caregivers) and health information (patients only) were collected

at study entry (see S2 and S3 Tables), and the outcome measures were collected every two months. In addition to the outcome measures, participants completed the following questionnaires that are not reported in this manuscript: (i) retrospective questions about the MQOL-E and QOLLTI-F v3 domains (for further study of their psychometric properties), (ii) Canadian Health Care Evaluation Project Questionnaire (CANHELP) Lite (both patient and family caregiver versions) [52], (iii) Veterans RAND 12-item Health Survey (VR-12) [52,53], and (iv) resource use questions for the purposes of economic evaluation. Missing or unavailable data from the questionnaires, including demographics (sex and age) of health information (diagnoses, healthcare services, and date of death), was supplemented with data from health records and administrative databases (including data from February 12, 2016, to November 1, 2020). The healthcare organizations extracted the requested data and shared the data with the participant's study identifier with the research team only for participants who provided consent to share this information for the purposes of this study and who provided their healthcare number.

**Outcome measures.** The primary outcomes of our study were the QOL of the patients and their family caregivers. Patient QOL was measured based on the overall summary score of the MQOL-E [42]. The MQOL-E has 20 items (scaled from 0 to 10, with item-specific anchors) that cover the physical, psychological, existential, social, (feeling like a) burden, environment, cognition, and health care domains. Family caregiver QOL was measured based on the overall summary score of the QOLLTI-F v3, which has 16 items (scaled from 0 to 10, with item-specific anchors) that represent seven domains, including their environment, the state of the patient, their own state, outlook, relationships, quality of care, and financial worries [45]. Both MQOL-E and QOLLTI-F have an item that assesses global QOL; it is not included in the summary scores. For both the MQOL-E and QOLLTI-F v3, the domain scores are the average of the corresponding items, after reverse-coding where necessary, and the overall summary scores are the average of the domain scores. For all MQOL-E and QOLLTI-F v3 domain and summary scores, '0' represents the poorest QOL and '10' represents the best QOL. The domain scores of the MQOL-E and the QOLLTI-F v3 were analyzed as secondary outcomes. Following the scoring guidelines for the MQOL-E and QOLLTI-F v3, multilevel multiple imputation was used to create 10 data files with imputed item response data for participants who had missing data for one or more of the items prior to calculating the overall and domain scores (1.9% and 2.1% imputed data values for MQOL-E and QOLLTI-F v3, respectively).

**Statistical analysis.** We described the patient and family caregiver samples for the intervention and control groups at the start of the study (Time 0) using frequencies and percentages for the categorical variables and means and standard deviations for the continuous variables. We examined the effect of the intervention based on an intention-to-treat analysis [54,55]. To do so, we first graphically compared the primary outcomes of the intervention and control groups by depicting the participants' individual trajectories as well as the group-average linear changes over the seven measurement occasions (i.e., every two months, over one year). Having observed substantial inter-individual variability and non-linear trajectories, we subsequently applied longitudinal structural equation models (SEMs), with a level and

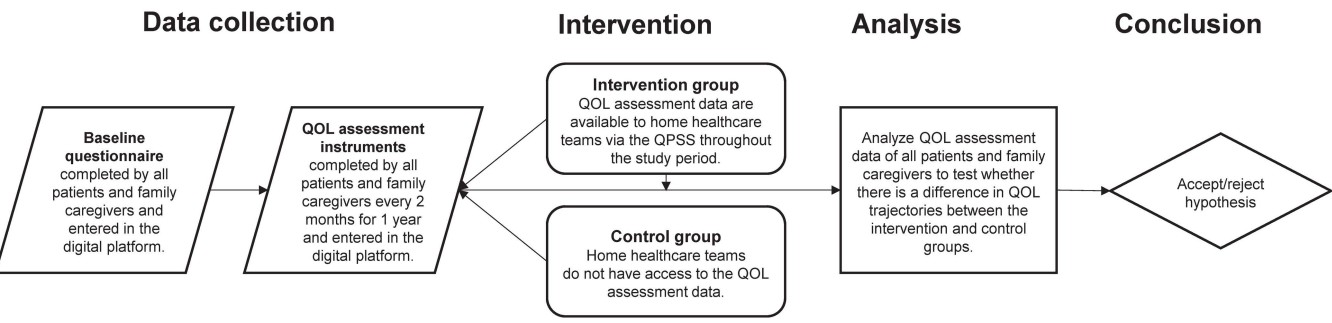

**Fig 3. Pragmatic Randomized Controlled Trial Design.**

shape (LS) specification, to statistically compare the average change over time of the intervention and control groups (Mplus software, version 8.6) [56,57]. We chose the LS model because of its flexibility in accommodating different patterns of change [58]. Further, the LS model allowed for direct examination of heterogeneity amongst the individuals' trajectories and the plausibility of the mean trajectories.

The SEMs were specified with two latent factors representing the intercept (i.e., the QOL score at the time the participant joined the study) and the slope (i.e., the average change in the QOL scores across all measurement occasions), which were regressed on the grouping variable (intervention vs. control). The regression coefficient for the intercept was fixed at zero (the groups were assumed to be equivalent due to random assignment) whereas the regression coefficient of the slope (intervention effect) was estimated. In accordance with LS, the loadings of the latent slope were estimated by fixing two of the time points: study entry was fixed at zero, and time point 7 (approximately one year after study entry) was fixed at one. The loadings of the latent slope for the other time points were allowed to vary (see Fig 4). To account for the complex sample structure, the SEMs for patients were clustered by home healthcare team. Global fit indices were used to guide our adjudication of model fit based on the following conventions: root mean square error of approximation (RMSEA) ≤.06, comparative fit index (CFI) ≤.95, and a standardized root mean squared residual (SRMR) ≤.08 [59]. In addition, residual correlations and modification indices were examined to identify variables and parameters with the greatest misfit.

To account for loss to follow up, we first explored the patterns of missingness over time and the reasons (i.e., death, no longer receiving homecare services, other reasons). We also compared the scores of the outcome at each measurement occasion for those who were lost to follow up at the subsequent timepoint and those who were not. Additionally,

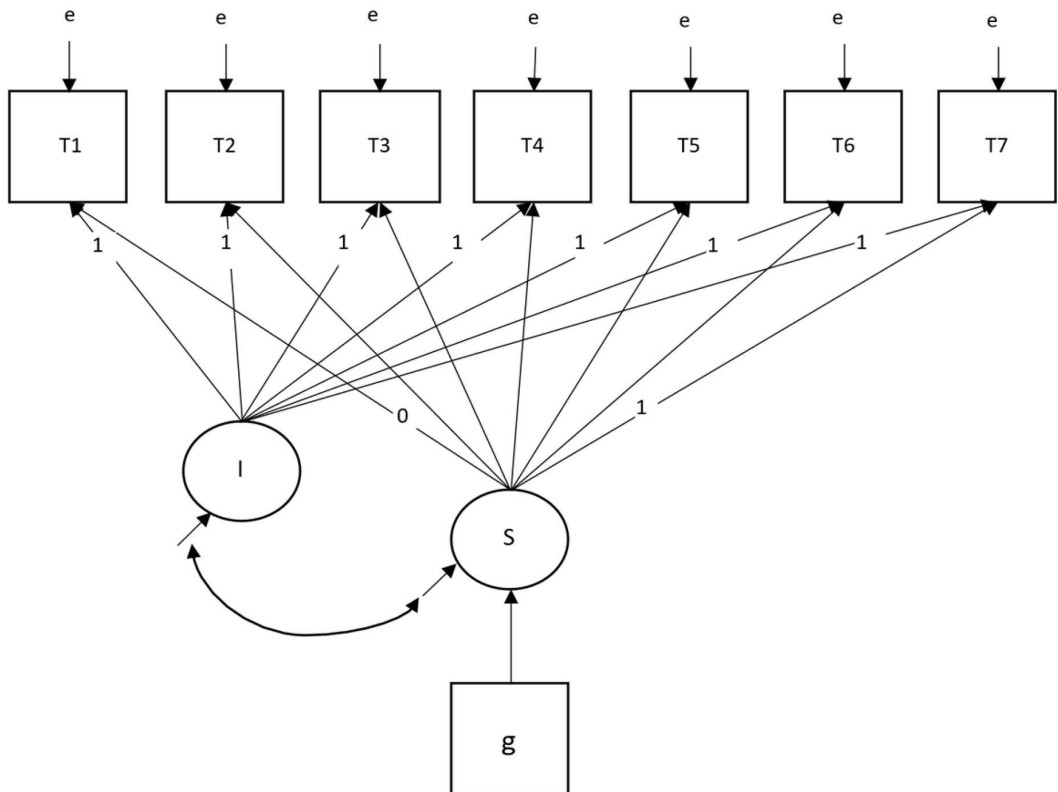

**Fig 4. Longitudinal Structural Equation Model. e = error variance. T1-T7 = time periods 1–7. I = latent intercept. S = latent slope. g = grouping variable.**

we explored whether the results differed for models with two to seven measurement occasions. We subsequently implemented the following procedures to ascertain whether study results pertaining to the intervention effect were influenced by different missing data assumptions: (i) full information maximum likelihood estimation, with the assumption that the data were missing at random, (ii) modeling reasons for loss to follow up as a missing data correlate, (iii) the Diggle-Kenward selection model, assuming the data were not missing at random, where loss to follow up was related to both past and current outcomes, and (iv) a pattern mixture model, assuming the data were not missing at random, where loss to follow up at different time points were modelled as covariates [60].

In addition to testing the hypotheses, we explored whether the slope of the QOL trajectories was associated with the intensity of clinicians' use of the QPSS. This was accomplished by regressing the slope of the SEMs on the number of

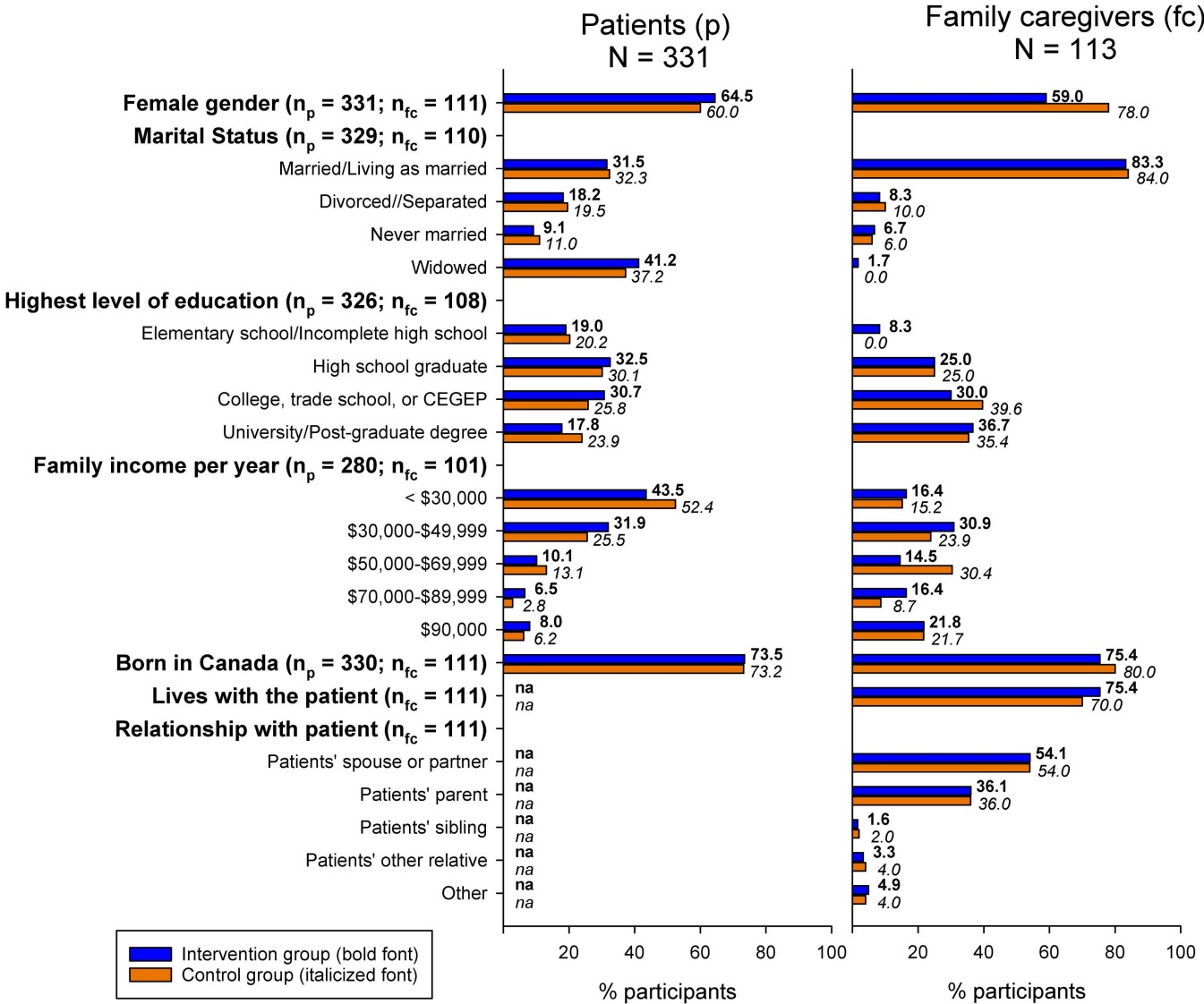

**Fig 5. Demographic Characteristics of Patients and Family Caregivers.** np = patient sample size. fc = family caregiver sample size. CECEP = general and professional teaching college in Quebec. na = not applicable. None of the differences between intervention and control groups had *p*-values <0.05. See S2 and S3 Tables for details.

occasions of QPSS use by home healthcare teams (which we defined as having viewed a patient's or family caregiver's QOL assessment data at least once on a given day) and the total duration of QPSS use (the time spent by home healthcare teams accessing the QPSS).

## Results

### Sample description

Three hundred seventy-three randomized patients were initially allocated to the intervention or control group, of which 154 provided follow-up QOL assessment data for one year and 219 were lost to follow up due to death (n = 40), no longer receiving home health care (n = 34), or for other reasons (e.g., worsening conditions, no reason provided, or no longer reachable; n = 145) (see Fig 1). The intention to treat analysis excluded 42 patients who only provided access to administrative data and did not complete any of the questionnaires, resulting in a total intention to treat sample of 331 (166 in the intervention group and 165 in the control group). About one third of the patients had a family caregiver who agreed to participate in the study (N = 113; 62 in the intervention group and 51 in the control group). Demographic characteristics of the patients and family caregivers were similar for the intervention and control groups (see Fig 5; detailed sample characteristics are provided in the Supplementary S2 and S3 Tables).

In the patient cohort, ages ranged from 55 to 100 years (mean = 79.1; SD = 9.9), 62.2% were women, and 31.9% were married or living as married, 39.2% were widowed, and 28.8% were divorced, separated or never married. Nearly one half of the patients (47.9%) had a family income of < $31,000 CAD and 7.1% had a family income of ≥ $90,000. Most patients identified their ethnic background as British (39.9%) or North American (14.9%), and nearly three quarters (73.3%) were born in Canada. The most common self-reported diagnoses, for which home health care was provided, included those pertaining to the nervous (26.8%), circulatory (21.6%) and musculoskeletal (21.6%) systems.

The family caregivers were between ages 29 and 93 years (mean = 66.5; SD = 13.5), 67.6% were women, and 83.6% were married or living as married, 0.9% were widowed, and 15.5% were divorced, separated or never married. Most of the family caregivers lived with the patient (73.0%), most were the patient's spouse or partner (54.1%) or the patient's daughter or son (36.0%), and a few were the patient's mother or father, another relative, or an employee (9.9%). The annual family income of the family caregivers ranged from the lowest category of < $31,000 (15.8%) to the highest category of ≥ $90,000 (21.8%). Most family caregivers identified their ethnic background as British (32.1%) or North American (15.6%), and 77.5% were born in Canada.

The pre-intervention (Stage 1) focus groups or one-to-one interviews were attended by 100 clinicians (94 clinicians participated in focus groups only, one in an interview only, and five in both). The Stage 2 focus groups included 38 clinicians, of which five also participated in interviews. Twenty of the 38 clinicians in Stage 2 had participated in Stage 1, giving a total of 118 clinicians across both stages. No clinicians withdrew during the course of the study. The majority of the clinicians were registered nurses (70.3%), were employed in a permanent full-time position (69.5%), had at least 21 years of professional experience (50%), and had been in their current position for at least 2.5 years (50%) (further details are provided in S4 Table).

The results below are structured according to action phases in the KTA cycle. Illustrative quotes for each qualitative theme (identified in italics below) are provided in S5 Table, along with examples of negative/alternative cases to represent the diversity of perspectives.

### Adapt, assess, and tailor the QPSS intervention to local contexts (KTA Action Phases B-D)

During the pre-intervention period, clinicians were involved in QPSS integration from design through to use. Each of the sites had 'champions,' leadership support, involvement, and *administrative buy-in of the QPSS* (see S5 Table, quote [q], q1-2), as well as one year of consultation to ensure that the information made available in the QPSS was tailored to *complement workflow and other data* routinely collected (q3-5). The clinicians believed that for the information to be used,

*data display* needed to be strategic: quickly searchable, displayed graphically, and, most importantly, printable (q6-8). Thus, the QPSS was programmed with a print or download function, which was available to all users (clinicians, patients, and family caregivers). The clinicians believed that a key barrier was *not having sufficient time, resources, or capacity* to look at or follow-up on needs identified in the QOL assessment data (q9-10), yet administrators encouraged their staff to use their available time and resources to follow up on the QOL information. After reviewing the preselection of QOL assessment instruments included in the QPSS (based on our prior research) [39], the clinicians were involved in determining the *appropriateness of the QOL assessment instruments;* they confirmed that the ESAS-r, the MQOL-E, QOLLTI-F v3 and the CSNAT were appropriate and would provide relevant information to inform the care provided by the home healthcare teams. The clinicians were also presented with the CANHELP Lite (both patient and family caregiver versions) [52] because it was part of the original QPSS prototype [39,40]. However, they recommended that the CANHELP scores should not be made available to the clinicians because of concern about patients and family caregivers not feeling safe to report their level of satisfaction with care to those who were providing the care. The CANHELP was therefore not included as part of the intervention, meaning that clinicians did not see participants' responses to the CANHELP. The clinicians further confirmed that there was no need for any other QOL assessment instruments to be included (q11).

Clinician feedback helped tailor how the information was presented and able to be searched, including: line graphs to represent change over time for each of the QOL assessment domains and items; bar graphs to compare the scores for each of the domains, side-by-side; and corresponding tables. In addition, the clinicians emphasized the importance of being able to view the QOL assessment data of the family caregivers linked to the respective patient. The home healthcare providers emphasized the importance of *clinical judgement regarding the use of QOL information* to inform their practice (q12-13). Negative case analysis of clinicians' perspectives highlighted remarkable consistency and did not reflect extreme divergent views.

**Use of the QPSS by home healthcare teams (KTA Action Phases E)**

Over the course of the intervention period (25 months), 48 home healthcare professionals viewed the QOL assessment data (i.e., a graph or tabulation) of a patient or family caregiver on 271 occasions (which we defined as having viewed a

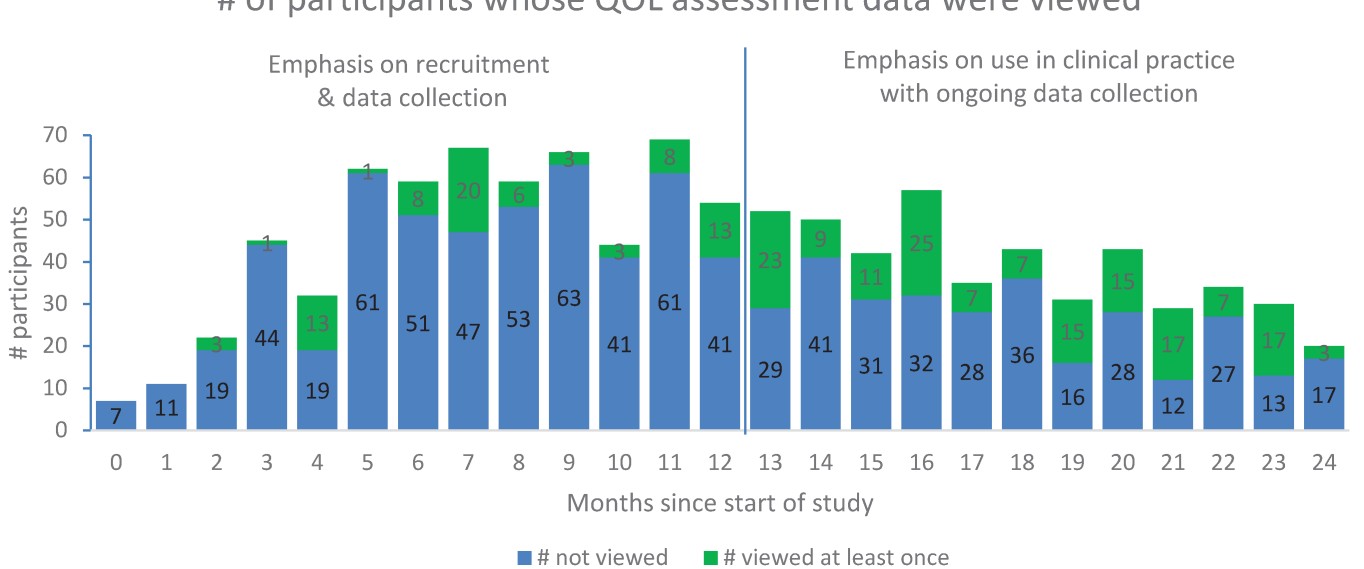

**Fig 6. Clinicians' Use of QPSS Over Time.**

patient's or family caregiver's QOL assessment data at least once on a given day). The median viewing duration for each occasion was 89 seconds (1st and 3rd quartiles=41 and 175 seconds). The clinicians viewed the QOL assessment data at least once throughout the duration of the project for 94 (56.6%) of the 166 patients and 27 (43.5%) of the 62 family caregivers in the intervention group. As Fig 6 shows, the clinicians infrequently, if ever, viewed the QOL assessment data provided by most of the patients and their family caregivers in the intervention group, throughout the study period, though there is some indication of greater use over time. The clinicians viewed the MQOL-E data from 90 (54.2%) of the 166 patients on 178 occasions (median duration=50 seconds per occasion), and the ESAS-r from 82 (49.4%) of the patients on 161 occasions (median duration=30 seconds) (see Fig 7). The clinicians' use of the QPSS for the family caregivers included 56 occasions for QOLLTI-F v3 data from 24 (38.7%) of the 62 family caregivers (median duration=55 seconds), and 50 occasions for CSNAT data from 22 (35.5%) family caregivers (median duration=24 seconds).

Overall, despite extensive engagement with the home healthcare teams at each site and initial enthusiasm, routine use of the QPSS by the home healthcare teams was limited; they did not frequently access the QPSS to review the available QOL information (see S5 Table, q14-15). However, there were some different experiences, including clinicians who accessed the QPSS more frequently and used the information to inform their care (negative case q16). Across the board, the clinicians deeply valued patients' perspectives and believed it was critical to their care. Clinicians explained that though they valued the QOL assessment information they did not access it regularly because of *competing priorities* (q17), yet there were a few examples of clinicians who saw use of QOL information as integral to the care they provided (negative case q18). A further hurdle was a reported lack of systems integration (i.e., having to go to an additional online site) (q19), yet a few clinicians also did not see this as a problem because it was on a different website (negative case q20). *Technological challenges* were a further deterrent from using the QPSS (q21). However, workarounds such as printing or viewing on a different device were found to support use of QPSS information in daily practice (negative case q22). Lastly, *being pressed for time* in one's practice was seen as an impediment (q23) yet not all held this view, rather seeing use of QPSS information as a resource to support holistic care (negative case q24).

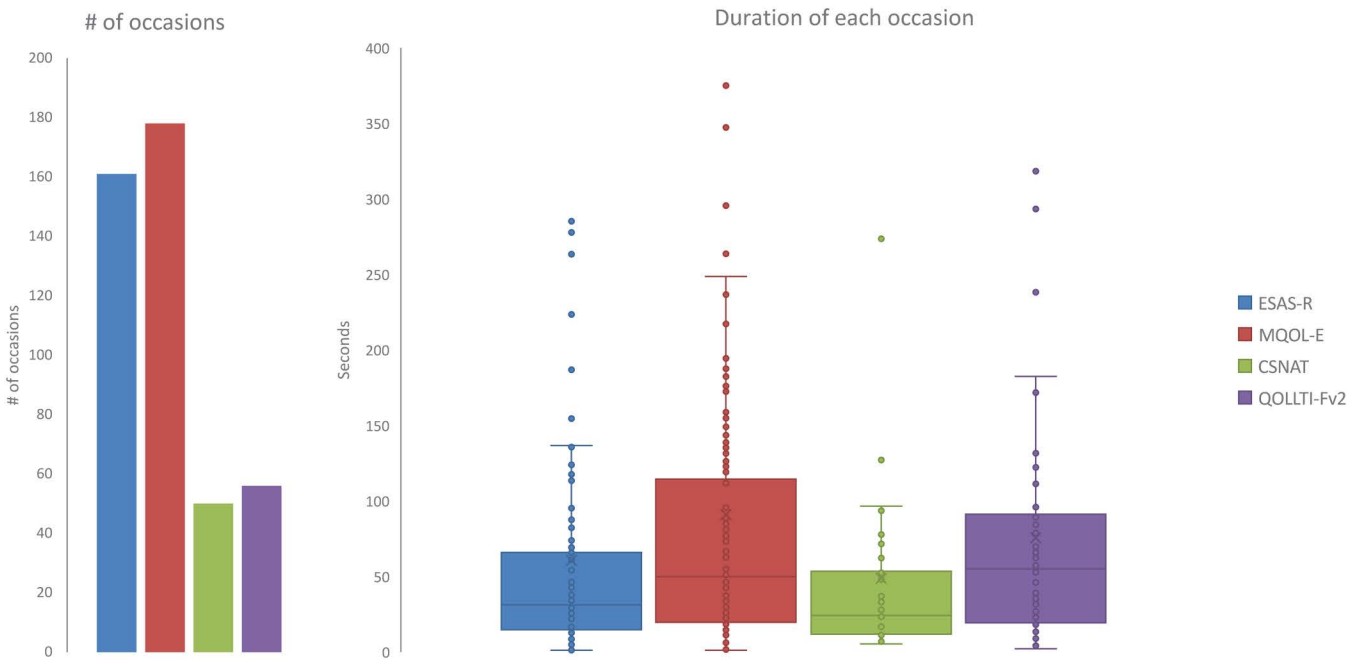

**Fig 7. QPSS Use Per Instrument.**

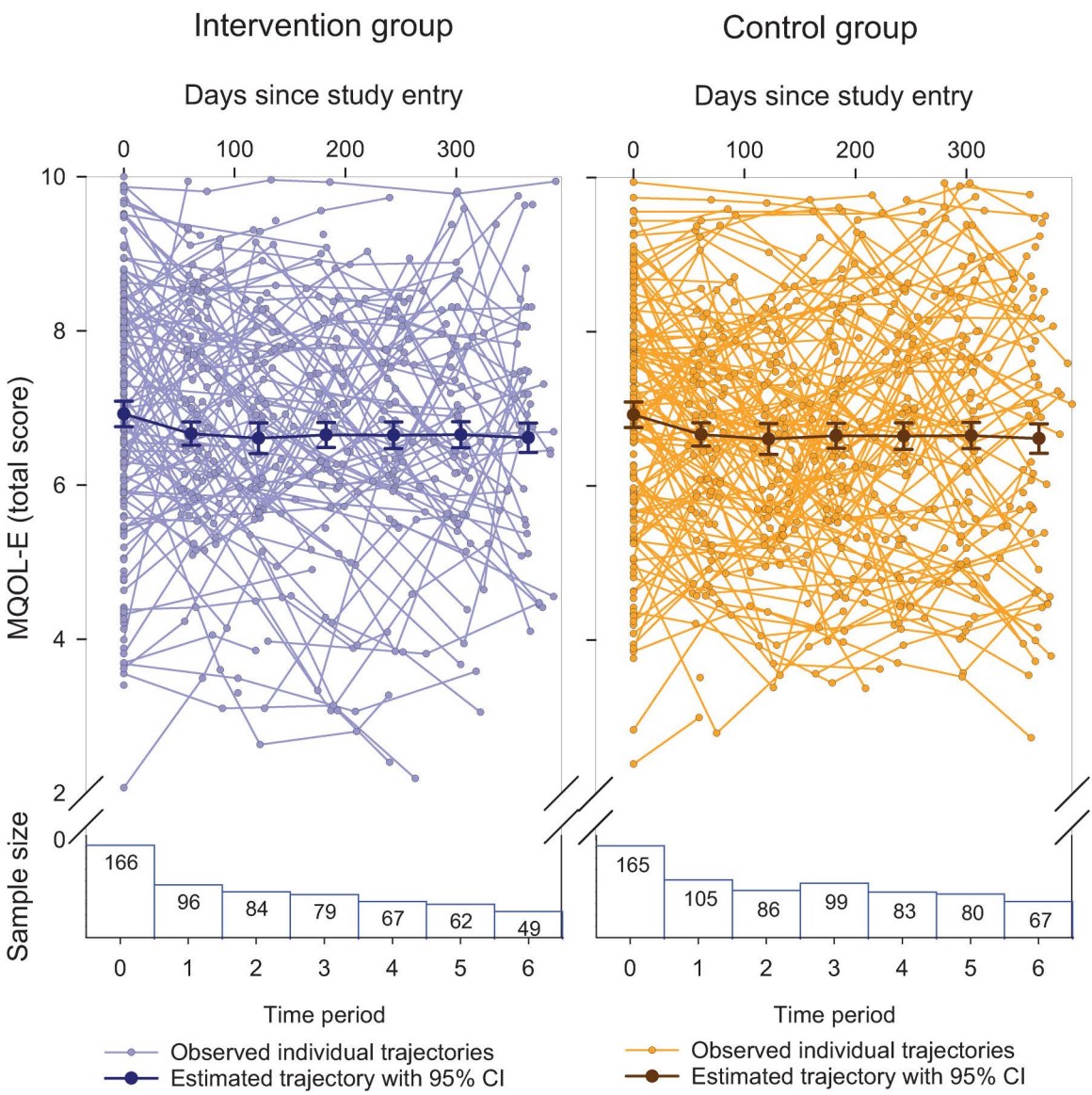

**Fig 8. MQOL-E Summary Score Trajectories.**

## Evaluation of patients' and family caregivers' QOL outcomes (KTA Action Phase F)

Of the 331 patients, 116 completed the final measurement occasion at one year (42 completed all measurement occasions and 74 had intermittent missingness). Comparisons of the missing data models indicated that the MAR model had the best fit, which was not improved with the inclusion of the reasons for loss to follow up as a missing data correlate. Although the Diggle-Kenward selection and pattern mixture models were more consistent with the pattern of missing data, these models resulted in a higher Bayesian Information Criterion. The *p*-values for the intervention effect for the various missing data models ranged from.18 to.53.

While Fig 8 indicates extensive individual-level variability in the trajectories of overall QOL, the estimated average trajectories were nearly flat and similar for the intervention and control groups. Accordingly, the longitudinal SEM results for patient QOL, based on the MAR model, indicated an estimated overall QOL score (based on the MQOL-E)

**Table 3. Structural Equation Model Results for Patients' QOL Trajectories.**

| | Physical Est. (SE) | Psycho-logical Est. (SE) | Social Est. (SE) | Exist-ential Est. (SE) | Environ-ment Est. (SE) | Cog-nition Est. (SE) | Health care Est. (SE) | Burden Est. (SE) | Overall QOL Est. (SE) |
|---|---|---|---|---|---|---|---|---|---|
| **Trajectories** | | | | | | | | | |
| Intercept | 5.15 (0.23) | 7.27 (0.17) | 7.67 (0.09) | 6.65 (0.13) | 8.15 (0.11) | 7.58 (0.06) | 7.55 (0.18) | 5.28 (0.22) | 6.92 (0.08) |
| Slope[a] | -0.02 (0.10) | -0.48 (0.24) | -0.35 (0.11) | -0.10 (0.15) | -0.67 (0.20) | -0.48 (0.13) | -0.15 (0.32) | -0.03 (0.11) | -0.31 (0.08) |
| Intervention effect[b] | 0.07 (0.19) | 0.21 (0.16) | -0.24 (0.24) | 0.00 (0.04) | 0.04 (0.21) | 0.28 (0.17) | 0.19 (0.36) | 0.08 (0.28) | -0.01 (0.13) |
| **Correlations & variances** | | | | | | | | | |
| $r$[c] | -0.09 (0.22) | 0.96 (0.95) | 0.00 (0.00)[d] | 0.00 (0.09) | -1.01 (1.94) | 3.15 (5.74) | 0.24 (0.76) | -0.03 (0.26) | -1.36 (2.16) |
| Var. intercept[d] | 3.02 (0.23) | 2.90 (0.88) | 3.08 (0.40) | 2.24 (0.33) | 3.23 (1.84) | -0.29 (5.81) | 2.28 (0.87) | 4.04 (0.74) | 2.93 (2.13) |
| Var. slope | 0.15 (0.43) | -0.71 (0.48) | 0.75 (0.36) | 0.04 (0.10) | 1.95 (2.22) | -2.85 (5.72) | -0.13 (0.41) | 0.09 (0.77) | 1.68 (2.15) |
| **Model fit** | | | | | | | | | |
| $X^2$[e] | 22.58 | 23.27 | 20.44 | 28.19 | 23.25 | 16.81 | 27.66 | 22.40 | 29.69 |
| RMSEA | 0.00 | 0.00 | 0.00 | 0.02 | 0.00 | 0.00 | 0.02 | 0.01 | 0.03 |
| CFI | 1.00 | 1.00 | 1.00 | 0.99 | 1.00 | 1.00 | 0.98 | 0.99 | 0.98 |
| SRMR | 0.05 | 0.08 | 0.06 | 0.05 | 0.07 | 0.04 | 0.08 | 0.05 | 0.10 |

Est. = parameter estimate. RMSEA = root mean square error of approximation. SRMR = standardized root mean squared residual. CFI = comparative fit index. Slope = change from time point 1–7. Intercept = score at time point 1.

[a]Average change across 7 measurement occasions (approximately 1 year).

[b]Unstandardized regression coefficients for the slope regressed on grouping variable (1 = control group, 0 = intervention group).

[c]$r$ = correlation of intercept and slope.

[d]Correlation between intercept and slope was fixed at zero for models with poor convergence and where the p-value of the initial estimate >0.05.

[e]Degrees of freedom = 24 for models with estimated intercept variance and 25 for models with correlation between intercept and slope fixed at zero.

of 6.92 at baseline, with a reduction in overall OQL, on average, over one year of 0.31 and 0.30 in the intervention and control groups, respectively (see Table 3). The difference in the slopes of the two groups (intervention effect) was thus extremely small (-0.01), with a *p*-value of.96. The distributions of the group-specific domain scores were similar at baseline and one-year follow up (see Fig 9). The SEM results confirmed that there was no substantial change over time in any of the domains, for both groups (see Table 3). The lowest (worst) average domain scores at baseline were for "physical" (5.15) and "feeling like a burden" (5.28), and the highest (best) was for "environment" (8.15). In addition, no associations were found when exploring the relationships between the slopes of the QOL trajectories (for overall QOL and each of the QOL domains) and frequency or duration of QPSS use by home healthcare teams.

Fig 10 reveals similar results for the overall QOL trajectories of the 113 family caregivers. The SEM results (see Table 4) indicated an estimated overall QOL score (based on the QOLLTI-F v.3) of 7.06 at baseline, with a reduction of 0.36 and 0.13 over one year in the intervention control groups, respectively. The intervention effect had a *p*-value of.34. The distributions of the group-specific domain scores were similar at baseline and one-year follow up (see Fig 9). The SEM results confirmed that there was no substantial change over time for any of the domains (see Table 4).

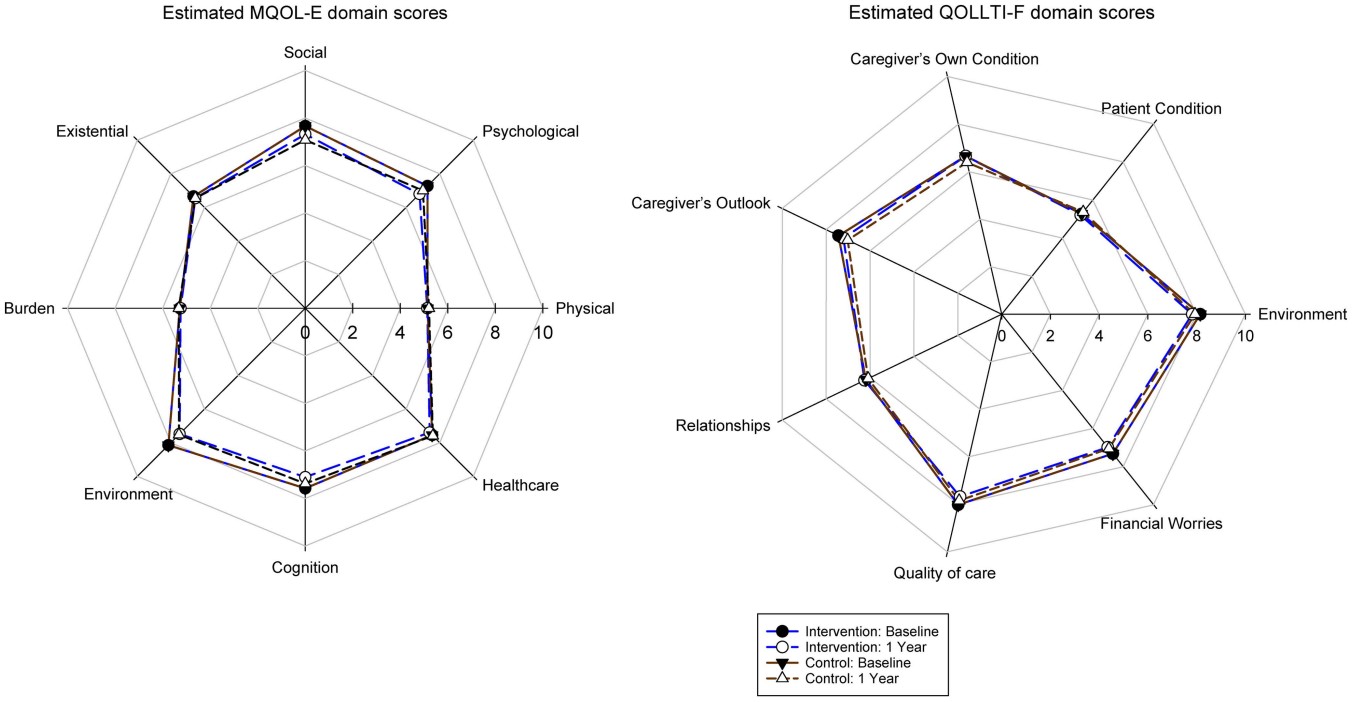

**Fig 9. MQOL-E and QOLLTI-F V3 Domain Scores at Study Entry and at One Year.**

## Discussion

Based on practice guidelines, previous studies, reviews about the use of QOL assessment tools (and PROMs) in clinical practice [20,61,62], and our discussions with healthcare professionals in the first stage of this study [20], we expected that the home healthcare teams would access data from the QPSS as part of their routine care. Accordingly, we hypothesized that having access to the QPSS over a period of one year (the intervention) would result in improved QOL for patients and their family caregivers. Our study results did not support this hypothesis (nor the expectation). The average trajectories for the intervention and control groups showed no change over time, and the trajectories were not found to be associated with the frequency or duration of QPSS use. Though the home healthcare teams expressed support for the intervention, they infrequently accessed the QOL information via the digital platform. Qualitative results identified competing priorities, a lack of systems integration, technological challenges, and time constraints as potential explanations for the clinicians' lack of use.

It is clear that the mere provision of access to QOL data via a digital platform for use by home healthcare teams is not sufficient to achieve improvements in QOL outcomes of patients and family caregivers. The question is: why? We offer four possible explanations. First, it is possible that an effect could not be detected because the home healthcare teams did not access the QPSS or sub-optimally used the QOL information (as indicated by the time spent). Similar challenges have been observed in other studies. An increasing body of implementation science studies points to a range of facilitators and barriers that may influence the implementation of digital QOL assessments (including electronic PROMs and PREMs) in healthcare settings [31]. For example, the initial enthusiasm but suboptimal use in our study resembles the implementation challenges and underutilization of the CSNAT in the Organising Support for Carers of Stroke Survivors (OSCARSS) trial, which was designed to improve family caregivers' outcomes [63,64]. Nonetheless, it is important to note that our study does not rule out the possibility that increased and optimized use by home healthcare teams and patients and family caregivers could have helped to better identify and address QOL concerns. Indeed, based on their systematic review of 11

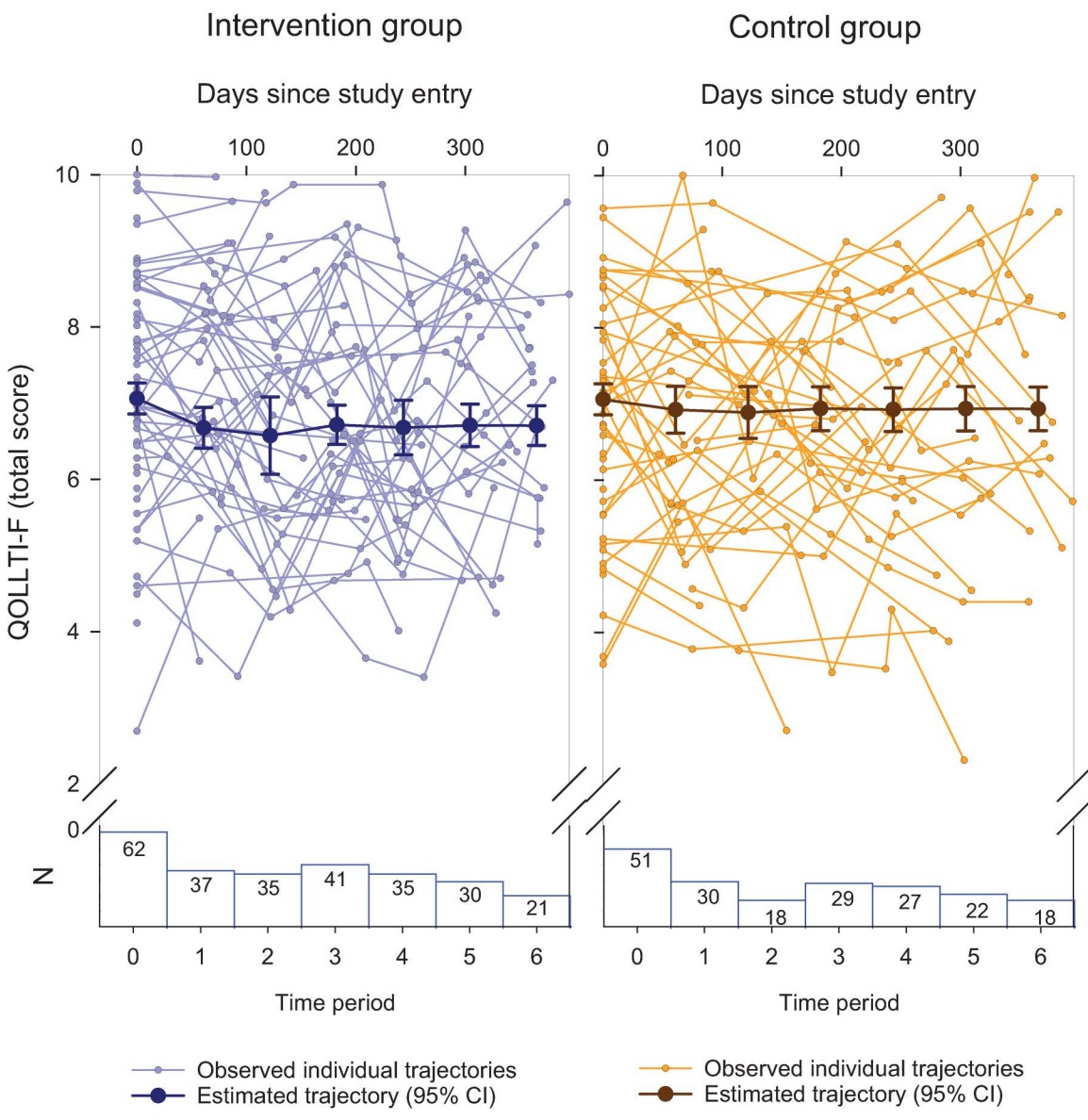

**Fig 10. QOLLTI-F v3 Total Trajectories.**

randomized controlled trials investigating the effects of 'PROM feedback' on patient's QOL, Gibbons et al. concluded that such an intervention "probably slightly improves quality of life" [20].

Second, some argue that optimized use of QOL assessment data by clinicians is a matter of better education, system integration, and resource provision [65]. An umbrella review designed to understand factors that influence routine integration of QOL assessments in clinical practice draws further attention to the need for "multicomponent organizational strategies covering training and guidance, necessary time and resources, roles and responsibilities, and consultation" [66]. Although formal education on the use of QOL information was not part of our study, all home healthcare teams received tailored, in-person instruction about accessing the QPSS and guidance on reviewing QOL information. We also provided home healthcare teams, patients and family caregivers with access to knowledge translation resources on "Quality of Life Assessments for

**Table 4. Structural Equation Model Results for Family Caregiver QOL Trajectories.**

| | Environ-ment Est. (SE) | Patient Est. (SE) | Own wellbeing Est. (SE) | Outlook Est. (SE) | Quality of care Est. (SE) | Rela-tional Est. (SE) | Financial Est. (SE) | Overall QOL Est. (SE) |
|---|---|---|---|---|---|---|---|---|
| **Trajectories** | | | | | | | | |
| Intercept | 8.15 (0.40) | 5.26 (0.41) | 6.65 (0.31) | 7.43 (0.14) | 8.02 (0.24) | 6.19 (0.12) | 7.32 (0.26) | 7.06 (0.10) |
| Slope[a] | -0.33 (0.50) | -0.06 (0.18) | 0.00 (0.45) | -0.22 (0.17) | -0.34 (0.20) | 0.06 (0.30) | -0.34 (0.51) | -0.36 (0.17) |
| Intervention effect[b] | 0.11 (0.64) | 0.16 (0.18) | -0.26 (0.32) | -0.19 (0.31) | 0.17 (0.31) | -0.17 (0.61) | 0.07 (0.61) | 0.23 (0.24) |
| **Correlations & variances[c]** | | | | | | | | |
| $r$[d] | 1.00 (0.63) | 1.63 (1.54) | 2.45 (0.23) | 2.40 (0.42) | 1.81 (0.69) | 1.91 (1.06) | 8.11 (1.61) | 1.46 (0.48) |
| Var. slope | -0.09 (1.00) | -0.88 (1.47) | -2.90 (0.65) | -2.60 (0.79) | -1.76 (1.02) | 0.48 (1.97) | -10.86 (2.67) | -1.43 (0.69) |
| **Model fit** | | | | | | | | |
| $X^2$ ($p$-value)[e] | 44.41 | 27.49 | 42.20 | 29.94 | 26.00 | 27.91 | 22.10 | 40.36 |
| RMSEA | 0.08 | 0.03 | 0.08 | 0.04 | 0.02 | 0.03 | 0.00 | 0.07 |
| CFI | 0.80 | 0.95 | 0.91 | 0.97 | 0.97 | 0.96 | 1.00 | 0.92 |
| SRMR | 0.12 | 0.13 | 0.14 | 0.18 | 0.14 | 0.11 | 0.08 | 0.14 |

Est. = parameter estimate. RMSEA = root mean square error of approximation. SRMR = standardized root mean squared residual. CFI = comparative fit index. Slope = change from time point 1–7. Intercept = score at time point 1.

[a]Average change across 7 measurement occasions (approximately 1 year).

[b]1 = control group and 0 = intervention group.

[c]Intercept variance was fixed at zero for all models.

[d]$r$ = correlation of intercept and slope.

[e]Degrees of freedom = 25 for all models.

Older Adults and Family Caregivers" (https:www.healthyqol.com/older-adults), including pamphlets and short videos, which were developed by our team as part of another project [67]. In addition, the QPSS was developed in close collaboration with clinical teams and integrated into a digital health information system [39], and the home healthcare teams were encouraged by their management to use the QOL assessment information to inform their care. Our overall approach was consistent with recommendations for planning, selection of measurement tools, and stakeholder engagement when PROMs are integrated with digital health systems [68]. Though it is possible that formal education on the use of QOL information may have been beneficial, evidence regarding the effectiveness of such educational interventions remains inconclusive [69].

Third, the lack of an intervention effect may have resulted from methodological challenges, especially with respect to measurement, the heterogeneity in individuals' QOL trajectories [70], and our reliance on conventional variable-centered statistical methods [71]. In general, standardized QOL assessment tools, involving fixed-form questionnaires, are based on the premise that people are consistent in how they interpret and respond to questions, that this consistency is maintained when using the tool to measure QOL over time, and that the tool includes the questions that are most relevant. It is, however, possible that some questions have different meanings for different people. It is also possible that people adapt as their condition progresses, which could result in a change in the meaning of their QOL over time; a phenomenon known as response shift [72]. If unaccounted for, such

inconsistencies between people and over time in the meaning of the questions about their QOL may result in heterogeneous QOL trajectories.

Fourth, in real-world research, there are many other factors that may result in heterogeneous QOL trajectories. The results of our study point to substantial heterogeneity in the QOL trajectories (as shown in the 'spaghetti plots' (Figs 8 and 10), which may be equated with heterogeneous treatment effects, undermining the conventional assumption that the intervention effect is the same for all people [73,74]. Additionally, loss to follow up was substantial in our study, which reduced the ability to accurately estimate average QOL trajectories within each group, thereby further obscuring the detection of an intervention effect.

The above methodological challenges raise concerns about the conventional use of variable-centered statistical methods to test for intervention effects on QOL trajectories of older adults who are living at home with chronic illnesses [75]. Analytical approaches that account for individual differences in QOL trajectories are recommended to disentangle heterogeneity in QOL trajectories [71]. People-centered measurement approaches using mixture computerized adaptive testing systems for tailoring the selection and scoring of questions to individuals may further help to reduce heterogeneity in QOL scores associated with potential measurement biases [76]. Additionally, we recommend further research focused on explaining heterogeneity in QOL trajectories using, for example, latent class models, growth mixture models, and other person-centred analytical methods and designs [71,75]. Further, realist evaluation strategies can help to examine why, for whom, and in what circumstances digital QOL assessments may be warranted [77].

## Conclusion

The results of our study do not confirm the hypothesis that providing home healthcare teams with access to QOL data obtained via standardized questionnaires in a co-designed digital QPSS will lead to improvements in QOL for patients and family caregivers. This may be due, in part, to lack of integration of the QPSS into routine practice and methodological challenges resulting in substantial individual variability in QOL trajectories. Future studies should focus on how QOL assessment data from patients and family caregivers could be best obtained and used by these persons themselves alongside home healthcare teams. Other person-centered approaches to QOL, including a focus on the priorities and concerns identified by patients and their family caregivers, and methods for evaluating heterogeneous QOL outcomes should also be explored.

## Supporting information

**S1 Checklist. CONSORT Checklist** .
(DOCX)

**S2 Table. Patient sample description.**
(DOCX)

**S3 Table. Family caregiver sample description.**
(DOCX)

**S4 Table. Sample description of qualitative clinician participants.**
(DOCX)

**S5 Table. Themes and illustrative quotes.**
(DOCX)

**S1 Protocol. QPSS Study Protocol.** .
(PDF)

## Acknowledgments

We are grateful to our Innovation Community, including researchers, healthcare providers, and patient partners who were part of the original project proposal (https://webapps.cihr-irsc.gc.ca/decisions/p/project_details.html?applId=334627) and subsequent implementation of the project, the people involved in the prior projects that informed the development of this project see: [39,40,67] & https://www.healthyqol.com/older-adults), and the research participants and home healthcare teams who contributed their time and freely shared their experiences by completing questionnaires and participating in interviews and focus groups.

## Author contributions

**Conceptualization:** Richard Sawatzky, Kara Schick-Makaroff, Pamela A. Ratner, Joakim Öhlén, David G. T. Whitehurst, Alies Maybee, Kelli Stajduhar, S. Robin Cohen.

**Data curation:** Richard Sawatzky.

**Formal analysis:** Richard Sawatzky, Kara Schick-Makaroff, Joakim Öhlén, Jae-Yung Kwon, Kelli Stajduhar, Pamela A. Ratner, S. Robin Cohen.

**Funding acquisition:** Richard Sawatzky, Kara Schick-Makaroff, Pamela A. Ratner, Joakim Öhlén, Alies Maybee, Kelli Stajduhar, Lisa Zetes-Zanatta, S. Robin Cohen, David G. T. Whitehurst.

**Investigation:** Richard Sawatzky, Kara Schick-Makaroff, David G. T. Whitehurst, S. Robin Cohen.

**Methodology:** Richard Sawatzky, Kara Schick-Makaroff, Pamela A. Ratner, Joakim Öhlén, David G. T. Whitehurst, S. Robin Cohen.

**Project administration:** Richard Sawatzky, Kara Schick-Makaroff.

**Resources:** Lisa Zetes-Zanatta.

**Supervision:** Richard Sawatzky, Kara Schick-Makaroff.

**Visualization:** Richard Sawatzky, Kara Schick-Makaroff.

**Writing – original draft:** Richard Sawatzky, Kara Schick-Makaroff, Pamela A. Ratner.

**Writing – review & editing:** Richard Sawatzky, Kara Schick-Makaroff, Pamela A. Ratner, Joakim Öhlén, Jae-Yung Kwon, David G. T. Whitehurst, Alies Maybee, Kelli Stajduhar, Lisa Zetes-Zanatta, S. Robin Cohen.

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
