## [Decision Letter · Decision Letter 0]

23 Sep 2024

PONE-D-24-23372Did a digital quality of life (QOL) assessment and practice support system in home health care improve the QOL of older adults living with life-limiting conditions and of their family caregivers? A mixed-methods pragmatic randomized controlled trial.PLOS ONE

Dear Dr. Sawatzky,

Thank you for submitting your manuscript to PLOS ONE. After careful consideration, we feel that it has merit but does not fully meet PLOS ONE’s publication criteria as it currently stands. Therefore, we invite you to submit a revised version of the manuscript that addresses the points raised during the review process.

**ACADEMIC EDITOR: **Many thanks for submitting your manuscript. As shown by the reviewers' comments, there is a need for substantial revisions. We hope you will be able to attend to these thoroughly. 

We look forward to receiving your revised manuscript.

Kind regards,

JONATHAN BAYUO, PhD

Academic Editor

PLOS ONE

Journal Requirements:

Reviewers' comments:

Reviewer's Responses to Questions

**Comments to the Author**

1. Is the manuscript technically sound, and do the data support the conclusions?

Reviewer #1: Yes

Reviewer #2: Yes

Reviewer #3: Yes

2. Has the statistical analysis been performed appropriately and rigorously? 

Reviewer #1: No

Reviewer #2: Yes

Reviewer #3: Yes

3. Have the authors made all data underlying the findings in their manuscript fully available?

Reviewer #1: Yes

Reviewer #2: Yes

Reviewer #3: Yes

4. Is the manuscript presented in an intelligible fashion and written in standard English?

Reviewer #1: Yes

Reviewer #2: Yes

Reviewer #3: Yes

5. Review Comments to the Author

Reviewer #1: What are the benefits of using SEM over linear mixed models (LMM) or Generalized Estimating Equations (GEE), which are generally easier to understand and interpret? LMM directly estimates group means and slopes and GEE provides flexibility in handling violations of normality.

Add more demographic /baseline information, e.g. Age, and chronic conditions.

Figure 4: Allocation (n=???

Reviewer #2: This is a very important paper about use of PROMs by home care teams. This is such a through report on methods and processes. Typically the reporting of methods such that interverventions cannot be replicated based on report is the primary critique in reviews. Thank you for the superior description of the intervention.

On page 21, you end with "Negative case analysis revealed that there were alternative perspectives to each of these explanations." You really leave your reader hanging. Can you provide some continuity?

Findings again are well reported

Discussion.

I would like to see more discussion about the use of the Carer Support Needs Assessment Tool Intervention. It is NOT meant to be used as a scale or checklist but as a conversation starter. The mean short scan of 24 seconds is disappointing. It was used in a study of changing needs over time in the transplant literature and certainly did show how needs changed over time. I suspect there may be a worthwhile paper on that from this research.

The other piece that I think where more discussion would be useful is on diversity -- caregivers and clients (spagetti plots). Discussion might be fruitful on the lack of person-centered care because with such diversity, in my view care needs to be person-centered in order to have any impact on outcomes such as quality of life or anxiety/ burden. Another area that would be useful to understand is why the cursory use of the added information. Were they able to tailor services based on the information in the assessments? What incentives did they have to use the new information?

All in all this is a paper that needs to be published. I would just like as fulsome a discussion as you had with methods. To me, there was not enough of a so-what to improve home care in the discussion.

This is excellent research and an excellent report.

Reviewer #3: Discuss strategies that could be employed to minimize loss to follow-up. What proactive measures were taken to maintain participant engagement throughout the study?

Can the authors clarify their choice of statistical methods for analyzing QOL data? Were any alternative methods considered that could account for heterogeneity in QOL trajectories?

Consider providing a more explicit summary of the key findings at the beginning of the discussion. This would help orient readers before delving into interpretations and implications.

You mention clinician engagement with the QPSS was limited. What strategies do you propose to enhance this engagement in future studies?

Have you considered external factors (e.g., changes in healthcare policies, community resources) that may have influenced the QOL outcomes during the study? How might these be controlled for in future research?

Consider highlighting specific recommendations for practice based on the findings.

6. PLOS authors have the option to publish the peer review history of their article (what does this mean? ). If published, this will include your full peer review and any attached files.

**Do you want your identity to be public for this peer review?** For information about this choice, including consent withdrawal, please see our Privacy Policy .

Reviewer #1: No

Reviewer #2: No

Reviewer #3: No

---

## [Author Response · Author response to Decision Letter 1]

3 Jan 2025

Please see the response to reviewers document.

---

## [Decision Letter · Decision Letter 1]

11 Feb 2025

PONE-D-24-23372R1Did a digital quality of life (QOL) assessment and practice support system in home health care improve the QOL of older adults living with life-limiting conditions and of their family caregivers? A mixed-methods pragmatic randomized controlled trial.PLOS ONE

Dear Dr. Sawatzky,

Thank you for submitting your manuscript to PLOS ONE. After careful consideration, we feel that it has merit but does not fully meet PLOS ONE’s publication criteria as it currently stands. Therefore, we invite you to submit a revised version of the manuscript that addresses the points raised during the review process.

We look forward to receiving your revised manuscript.

Kind regards,

JONATHAN BAYUO, PhD

Academic Editor

PLOS ONE

Journal Requirements:

Reviewers' comments:

Reviewer's Responses to Questions

**Comments to the Author**

1. If the authors have adequately addressed your comments raised in a previous round of review and you feel that this manuscript is now acceptable for publication, you may indicate that here to bypass the “Comments to the Author” section, enter your conflict of interest statement in the “Confidential to Editor” section, and submit your "Accept" recommendation.

Reviewer #1: (No Response)

Reviewer #2: All comments have been addressed

2. Is the manuscript technically sound, and do the data support the conclusions?

Reviewer #1: (No Response)

Reviewer #2: Yes

3. Has the statistical analysis been performed appropriately and rigorously? 

Reviewer #1: (No Response)

Reviewer #2: Yes

4. Have the authors made all data underlying the findings in their manuscript fully available?

Reviewer #1: (No Response)

Reviewer #2: Yes

5. Is the manuscript presented in an intelligible fashion and written in standard English?

Reviewer #1: (No Response)

Reviewer #2: Yes

6. Review Comments to the Author

Reviewer #1: LMM and GEE are simpler and more flexible approaches for handling repeated measures. You can treat time points as categorical variables, reducing the need to specify a linear or nonlinear parametric pattern. Individual variability can also be explained with variance-covariance matrix setup. Furthermore, mixed models provide unbiased estimations when data are missing at random, and can handle different missing pattern with imputation.

Reviewer #2: I cannot wait to cite this paper. It is excellent. Thank you. In our work and other work with family caregivers, health providers always rate themselves as very good or excellent in their communication and partnership with family caregivers. The conundrum from Miichelle Lobchuk's "In your shoes" communication work that rarely do family caregivers give health providers more than 50% in their first ratings.

7. PLOS authors have the option to publish the peer review history of their article (what does this mean? ). If published, this will include your full peer review and any attached files.

**Do you want your identity to be public for this peer review?** For information about this choice, including consent withdrawal, please see our Privacy Policy .

Reviewer #1: No

Reviewer #2: No

---

## [Author Response · Author response to Decision Letter 2]

14 Feb 2025

See cover letter and response to reviewers.

---

## [Decision Letter · Decision Letter 2]

18 Feb 2025

Did a digital quality of life (QOL) assessment and practice support system in home health care improve the QOL of older adults living with life-limiting conditions and of their family caregivers? A mixed-methods pragmatic randomized controlled trial.

PONE-D-24-23372R2

Dear Dr. Sawatzky,

We’re pleased to inform you that your manuscript has been judged scientifically suitable for publication and will be formally accepted for publication once it meets all outstanding technical requirements.

Kind regards,

JONATHAN BAYUO, PhD

Academic Editor

PLOS ONE

Additional Editor Comments (optional):

Reviewers' comments:

Reviewer's Responses to Questions

**Comments to the Author**

1. If the authors have adequately addressed your comments raised in a previous round of review and you feel that this manuscript is now acceptable for publication, you may indicate that here to bypass the “Comments to the Author” section, enter your conflict of interest statement in the “Confidential to Editor” section, and submit your "Accept" recommendation.

Reviewer #1: All comments have been addressed

2. Is the manuscript technically sound, and do the data support the conclusions?

Reviewer #1: (No Response)

3. Has the statistical analysis been performed appropriately and rigorously? 

Reviewer #1: (No Response)

4. Have the authors made all data underlying the findings in their manuscript fully available?

Reviewer #1: (No Response)

5. Is the manuscript presented in an intelligible fashion and written in standard English?

Reviewer #1: (No Response)

6. Review Comments to the Author

Reviewer #1: All concerns are addressed.

7. PLOS authors have the option to publish the peer review history of their article (what does this mean? ). If published, this will include your full peer review and any attached files.

**Do you want your identity to be public for this peer review?** For information about this choice, including consent withdrawal, please see our Privacy Policy .

Reviewer #1: No

---

## [Editor Report · Acceptance letter]

PONE-D-24-23372R2

PLOS ONE

Dear Dr. Sawatzky,

I'm pleased to inform you that your manuscript has been deemed suitable for publication in PLOS ONE. Congratulations! Your manuscript is now being handed over to our production team.

Kind regards,

on behalf of

Dr. JONATHAN BAYUO

Academic Editor

PLOS ONE